

## Projections of actual and potential evapotranspiration from downscaled high-resolution CMIP6 climate simulations in Australia

Hong Zhang[1], Sarah Chapman[1,2], Ralph Trancoso[1,2], Rohan Eccles[1], Jozef Syktus[2], Nathan Toombs[1]

1. Climate Projections and Services, Queensland Treasury, Queensland Government, GPO Box 2454, Brisbane QLD 4001, Australia

2. School of the Environment, The University of Queensland, Brisbane QLD 4001, Australia

*Correspondence to*: Hong Zhang (hong.zhang@des.qld.gov.au)

**Abstract**. Quantifying the impact of climate change on actual and potential evapotranspiration (AET and PET) is essential for water security, agriculture production and environmental management. AET and PET are strongly influenced by local factors such as topography, land cover and soil moisture, which limits the usability of global climate models for their projections. Here, we dynamically downscale CMIP6 models using CCAM to a 10km resolution over Australia and derive AET and PET at a daily time step using the Morton method and project future changes under SSP126, 245 and 370.

The performance of observation- and downscaled climate model-based AET is assessed against measured AET from 26 OzFlux sites in Australia. We show that high resolution downscaled climate models can provide reasonably accurate estimations of AET in Australia, with an ensemble mean error of 17% for historical period 1981-2010. This compared favourably to observation- and reanalysis-based products, which reported mean errors ranging from 15.7% – 44%.

Annual average end-of-century AET projections for low and intermediate emission scenarios (SSP126 and SSP245) show a decrease of -4.5% and -3.5% respectively in Australia, while under high emissions (SSP370) AET was projected to increase by 1.8%. In contrast, PET was projected to increase by 5.0% for SSP126, 8.4% for SSP245 and 11.5% for SSP370.

Using random forest model, we show that the primary controlling factors for changes in AET are precipitation and solar radiation, and solar radiation and maximum temperature for PET. Our results offer new insights into future AET and PET changes estimated using downscaled CMIP6 simulations with implications for agriculture, water supply and natural resources management.

## 1 Introduction

The combined processes of evaporation and transpiration, known as evapotranspiration, play a key role in the water cycle. Actual evapotranspiration (AET) is a collective term for the transfer of water to the atmosphere from both vegetated and unvegetated land surfaces. It is a large component of the water balance, and it has been estimated that land surface AET returns about 60% of annual precipitation into the atmosphere globally (Oki and Kanae, 2006). AET is widely used in environmental, agricultural, and water resource management applications including hydrology, irrigation, forestry and drought forecasting. AET is controlled by many factors such as vegetation type, soil type and moisture, irrigation and drainage, meteorological conditions, and water management in croplands (Allen et al., 1998). The accurate calculation of AET is important for crop water management, irrigation planning, water balance estimates, climate characterization and climate change studies (Glenn et al., 2011; Han and Tian, 2020; Wanniarachchi and Sarukkalige, 2022; Ghiat et al., 2021). Climate change is expected to change AET in the future, as temperatures rise, and precipitation patterns change. This will impact the Earth's water cycle, including the frequency and intensity of floods and drought. Therefore, understanding how climate change will impact AET is imperative if we are to understand changes to the hydrology cycle.



In Australia, the driest inhabited continent in the world, it is crucial to understand future changes in
      AET to build resilience to climate change. The largest consumptive component of the water budget is
      the water that's removed from the land-surface and plants by AET, and almost 90 per cent of the
      precipitation that falls on the Australian continent is returned through AET to the atmosphere (Bureau
      of Meteorology, 1988). Since 2010 the station observational network of pan evaporation has been in
decline, with an increasing reliance on AET monitoring tools based on satellite technology. For
      example, water use dynamics can now be tracked from paddock-to-continental scales thanks to
      advanced tools such as the AET explorer (TERN, 2024), which leverages remotely sensed datasets.
      Australia is unique in climate and biodiversity due to nutrient-poor soils, high climatic variability, high
      fire frequencies and a generally flat topography. Rain and potential evapotranspiration (PET) are the
main drivers of yield for cropping (Barrett-Lennard et al., 2024). Thus, accurate evapotranspiration
      modelling is required to sustainably manage Australian agriculture and water resources.

      AET is one of the most important water balance variables. It is widely held to be the most difficult
      water balance component to directly measure because significant care is required in setting up and
      processing data (Allen et al., 2011; Wang and Dickinson, 2012). Although eddy covariance towers are
considered among the most reliable systems to estimate AET, their use is limited due to high
      installation and maintenance costs. Given these complexities, long-term and robust AET
      measurements are scarce. Therefore, models which predict AET and PET are widely used due to their
      relative simplicity and due to the availability of meteorological data. Understanding future AET is
      hampered by the difficulty in measuring it, when compared to other hydrological values such as
precipitation and streamflow (Yang, 2023).

      Given the difficulty in measuring AET directly, many models have been developed to estimate it,
      including: a) water balance (Falalakis and Gemitzi, 2020; Guerschman et al., 2022); b) energy balance
      (Crago et al., 2022; Kim and Kaluarachchi, 2018; Ma et al., 2015; McMahon et al., 2016; Pan et al.,
      2024; Taheri et al., 2022; Tu et al., 2023; Xu and Singh, 2005; Yang et al., 2011); c) data driven methods
(Glenn et al., 2011; Guo et al., 2024; Karahan et al., 2024) and d) semi-empirical and empirical methods
      (Priestley and Taylor, 1972; Raja et al., 2024). A number of new techniques based on the
      complementary relationship (CR) have been developed recently (Crago et al., 2022; Kim and
      Kaluarachchi, 2018; Liu et al., 2022a; Ma et al., 2015; Shang et al., 2022; Tu et al., 2023; Zhang et al.,
      2017). CR provides a framework for estimating AET with routine meteorological data by
acknowledging the relationship between AET and PET. The performance of CR models has also been
      investigated (Ma et al., 2015; Mobilia and Longobardi, 2021; Shang et al., 2022; Zhang et al., 2017)
      and in some regions it has not been possible to identify a single model which outperforms the others
      (e.g., Mobilia and Longobardi, 2021). In data-scarce regions, the nonlinear-CR model is recommended
      in the absence of measured AET data for local calibration of model parameters (Ma et al., 2015).

AET estimates rely on several types of climate data as input (e.g., solar radiation, vapour pressure, and
      temperature). For historical estimates, global and regional climate model outputs or observations,
      such as ground based, or remotely sensed data can be used as inputs. For future projections, global
      and regional climate models are the preferred methods to estimate AET along with trend
      extrapolation and machine learning techniques. There are several historical AET sources available
globally and for Australia: Copernicus AET (Wit et al., 2022), GLEAM (Martens et al., 2017; Miralles et
      al., 2011), DOLCE (Hobeichi et al., 2018), ERA5-Land (Muñoz-Sabater et al., 2021), CMRSET (McVicar
      at al., 2022), SILO (Jeffrey et al., 2001) and AWO (Wilson et al., 2022). Most of them are derived from
      satellite observations, station-based interpolation, and reanalysis. Future projections using regional
      climate modelling are also available (Peter et al., 2023). Before using any of these products, however,
it is necessary to evaluate them against observed AET to determine how well they perform.

      Most validation techniques to evaluate the performance of AET products make use of point
      observations such as eddy-covariance flux towers (Guo et al., 2024; Mobilia and Longobardi, 2021; Tu
      et al., 2023). For example, Khan et al. (2020) compared observation-based AET datasets with five eddy-



covariance-based in-situ flux towers across heterogeneous Australian landscapes. Based on their
accuracy assessment, agreement between four observation-based AET datasets and the flux tower
measurements followed the order: AWRA-L > GLEAM > GLDAS > MOD16. Performance evaluations
have also been carried out for global remotely sensed AET products with GLEAM and AVHRR
outperforming other products at point scale (Zhu et al., 2022). However, it is still challenging to
determine the best AET product in all respects and at all scales. For example, Fuentes et al. (2024)
found that MODIS and Synthetic AET had the highest correlation skills at point and catchment scales,
whilst PML and TerraClimate AET had the lowest magnitude errors.

OzFlux data (Beringer et al., 2016) has been used to undertake micrometeorological measurements
to monitor exchanges of carbon, water vapour and energy across Australian ecosystems. Although
potential evaporation can be quantified from a spatial network of pan evaporation data dating back
to 1975 (Roderick and Farquhar, 2004), OzFlux sites provide the only observations of AET (Beringer et
al., 2022). OzFlux AET data were used in the evaluation of modelled AET in the operational Australian
Water Resources Assessment - Landscape model (AWRA-L) (Frost et al., 2015). They have also been
used to constrain large-scale AET estimates from process- and satellite-based models, yielding a data-
constrained estimate of mean Australian AET over the period 2000–2010 of $360 \pm 205$ mm yr$^{-1}$
(Hobeichi et al., 2021). Ozflux provide important ground-truth data for parameterizing, validating, and
improving satellite remote sensing and global inversion products. Here, we use OzFlux AET data to
validate gridded observation-based AET products as well as those estimated from regional climate
model outputs.

In addition to understanding present-day AET, understanding future AET change is important for
future water security, agriculture and vegetation productivity. Here, we use high-resolution climate
model simulations to estimate historical and future AET in Australia using Morton's energy balance
method (Morton, 1983). Morton models are widely used in hydrological applications because of their
flexibility and absence of wind speed in their formulation. Morton's AET model is popular because it
does not require soil water information and complex hydrological physics. The Queensland
government has previously produced high-resolution projections based on the CMIP5 models,
QldFCP-1 (Syktus et al., 2020a; Syktus et al., 2020b). These projections underpinned the development
of the Queensland Climate Adaptation Strategy 2017-2030, which includes an aim to improve
knowledge of climate change impacts in Queensland. Recently the datasets have been updated using
the latest CMIP6 models, which provide high-resolution projections for the whole continent at 10 km
spatial resolution. These new projections are especially important in the context of recent policy
developments in response to climate change, including the National Climate Resilience and
Adaptation Strategy (Department of Agriculture Water and the Environment, 2021), reforms to the
Safeguard Mechanism (Department of Climate Change, Energy, the Environment and Water, 2024b),
and the planned National Climate Risk Assessment (Department of Climate Change, Energy, the
Environment and Water, 2024a). This  new dataset is referred to as Queensland Future Climate
Projections - Phase 2 (QldFCP-2;Chapman et al., 2024), to distinguish it from QldFCP-1.

Climate models are required to evaluate the effects of climate change. However, before doing so we
need to evaluate their performance in the historical period to determine their robustness. Climate
model outputs usually have systematic biases that need to be corrected before their data can be
applied to impact assessment studies. Various bias correction methods have been developed to
improve the fitting between observations and simulations (Cannon et al., 2015; Dowdy, 2023; Zhang
et al., 2024). Most bias correction studies focus on precipitation and temperature, to the detriment of
evapotranspiration variables. Since site-based observation datasets such as OZFlux are not suitable to
perform bias correction on gridded data, there is a need to evaluate observation-based gridded
products against quality observational point data to benchmark bias correction. Thus, this study has
four main objectives:





i)    To evaluate the performance of AET estimates from observational-based products and from the QldFCP-2 dataset for historical period.

ii)   To bias correct the QldFCP-2 AET dataset based on the best performing AET observational product and assess the effects of bias correction on mean climate and projected changes.


iii)  To assess the climate change impacts on evapotranspiration in Australia across eight Natural Resource Management (NRM) regions.

iv)   To investigate the regional drivers for changes in AET and PET.

**2 Data and Study Area**

**2.1 Study region**

We evaluate observation-based and QldFCP-2 derived AET products over Australia (Figure 1). Australia provides a useful study area for estimating AET and evaluating model performance due to its diverse landscape and climate regions, including equatorial, tropical, sub-tropical, temperate, and arid

climate, and the presence of mountainous and coastal areas. We use the OzFlux data for evaluating AET products. The OzFlux network is distributed over different vegetation types (e.g., grassland, woodlands, forests, pasture, savannas, and wetland) and climate regions.

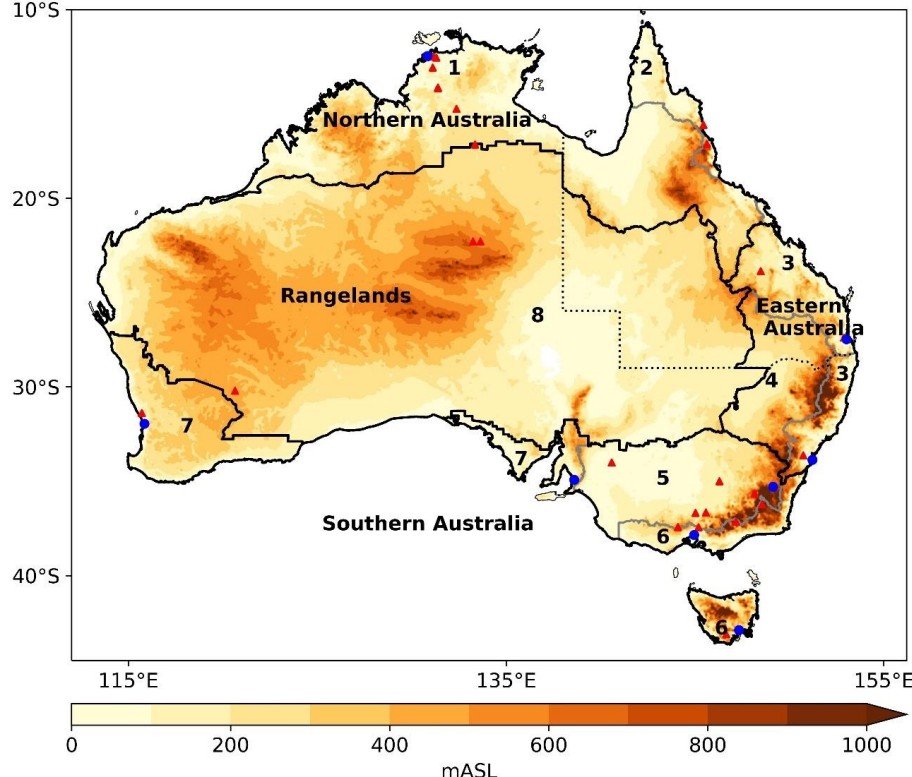

**Figure 1: Map showing domain of analysis and elevation, derived from SRTM (Shuttle Radar**

**Topography Mission (Gallant et al., 2009)). mASL = meters above sea level.  Triangles (in red)**



indicate the location of the 26 flux towers where AET was observed. Circles (in blue) indicate the location of the 8 capital cities of Australian states and territories. Natural Resource Management (NRM) regions include: 1-Monsoonal North (MN), 2-Wet Tropics (WT), 3-East Coast (EC), 4-Central Slopes (CS), 5-Murray Basin (MB), 6-Southern Slopes (SS), 7-Southern and South-Western Flatlands (SSWF), and 8-Rangelands (R) (CSIRO, 2020).

### 2.2 QldFCP-2 experimental setup

We dynamically downscale CMIP6 models using the Conformal Cubic Atmospheric Model (CCAM) developed by CSIRO (see Table 1 for list of CMIP6 host models), which is described in previous work (Chapman et al., 2023; 2024). As a result, a set of 60 simulations at a 10 km resolution are produced, including 15 simulations for the historical period and 15 simulations for each of three emission scenarios (SSP126, 245 and 370). Daily data for solar radiation, vapour pressure, maximum and minimum temperature, mean sea level pressure, precipitation, leaf area index and wind speed from QldFCP-2 are used to estimate AET and PET and to assess the primary drivers of future change to AET and PET.

**Table 1: CMIP6 GCMs and ensemble members selected for downscaling. Latitude and longitude grid size shown in resolution column. GCMs were downscaled using different CCAM configurations – atmospheric model only or atmosphere-ocean coupled.**

| CMIP6 Model | Model full name | Resolution | Ensemble member | Weighting | CCAM setup |
|---|---|---|---|---|---|
| ACCESS-ESM1.5 | Australian Community Climate and Earth System Simulator, version 1.5 | 1.875 x 1.25° | r6i1p1f1 | 0.33 | atmospheric |
| | | | r20i1p1f1 | 0.33 | atm-ocean coupled |
| | | | r40i1p1f1 | 0.33 | atm-ocean coupled |
| ACCESS_CM2 | Australian Community Climate and Earth System Simulator, version 2 | 1.875 x 1.25° | r2i1p1f1 | 1 | atm-ocean coupled |
| CMCC-ESM2 | Centro Euro-Mediterraneo sui Cambiamenti Climatici | 0.9 x 1.25° | r1i1p1f1 | 1 | atmospheric |
| CNRM-CM6-1-HR | Centre National de Recherches Météorologiques Coupled Global Climate Model, version 6.1, high-resolution | 0.5 x 0.5° | r1i1p1f2 | 0.5 | atmospheric |
| | | | r1i1p1f2 | 0.5 | atm-ocean coupled |
| EC-Earth3 | European Community Earth-System Model, version 3 | 0.8 x 0.8° | r1i1p1f1 | 1 | atmospheric |
| FGOALS-g3 | Flexible Global Ocean-Atmosphere-Land System Model, grid point version 3 | 2.5 x 2.5 | r4i1p1f1 | 1 | atmospheric |
| GFDL-ESM4 | Geophysical Fluid Dynamics Laboratory Earth System Model, version 4 | 1 x 1° | r1i1p1f1 | 1 | atmospheric |
| GISS-E2-2-G | Goddard Institute for Space Studies Model E2.2G | 2. x 2.5° | r2i1p1f2 | 1 | atmospheric |
| MPI-ESM1-2-LR | Max Planck Institute Earth System Model, version 1.2, low resolution | 1.9 x 1.9 | r9i1p1f1 | 1 | atmospheric |
| MRI-ESM2-0 | Meteorological Research Institute Earth System Model, version 2.0 | 1.125 x 1.125° | r1i1p1f1 | 1 | atmospheric |
| NorESM2-MM | Norwegian Earth System Model, version 2, 1 degree resolution | 1 x 1° | r1i1p1f1 | 0.5 | atmospheric |
| | | | r1i1p1f1 | 0.5 | atm-ocean coupled |

Note that 11 model ensemble mean is calculated by weighting models according to the number of host global climate models (Column 5 in Table 1 shows weightings). For example, because there are three ACCESS-ESM1.5 models, we would treat them together as being equal to one model with weight of 0.33. This approach was similarly applied to two CNRM-CM6-1-HR models and two NorESM2-MM models, respectively.

### 2.3 Calculations of potential and actual evapotranspiration



Evapotranspiration can be computed as PET or AET. PET relates to atmospheric demand and represents water loss from a hypothetical surface with unlimited water supply. AET accounts for landscape controls on evaporation and transpiration such as water availability and vegetation water use efficiency. We use the Morton's Complementary Relationship Areal Evapotranspiration (CRAE) Model outlined in Morton (1983) to calculate Point PET, Wet Environment Areal PET and Areal AET. We made minor modifications to the calculations to derive daily, rather than monthly estimates. PET represents evapotranspiration over a wet surface that is so small the local evapotranspiration would not affect the surrounding environment. The model assumes latent and sensible heat transfer near the surface only occurs via convection. AET represents evapotranspiration over a large area, given the prevailing soil-water conditions. The model assumes upwind effects are negligible and local variations are ignored, so the estimate is an areal average. As a surface undergoes drying from initially moist conditions, the PET increases while AET decreases. Morton's CRAE model uses this relationship between PET and AET to estimate the evapotranspiration from terrestrial surfaces.

In CRAE model, an equilibrium temperature ($T_p$ in °C), is defined as the temperature at which Morton's energy budget and mass transfer methods yield the same result for the PET rate of a moist surface.

$$PET = \frac{1}{\lambda}\left\{(R_n - G) - \left[\gamma f_t + 4\varepsilon\sigma(T_p + 273.15)^3\right](T_p - T_a)\right\} \tag{1}$$

$$PET = \frac{1}{\lambda}\left[f_t(e_p - e_a)\right] \tag{2}$$

where $R_n$ is net radiation (Wm$^{-2}$), $G$ is the soil heat flux into the ground surface (Wm$^{-2}$), $\gamma$ is the psychrometric constant (kPa °C$^{-1}$), $\lambda$ is the latent heat of vaporization (J kg$^{-1}$), and $f_t$ is the vapor transfer coefficient (Wm$^{-2}$ kPa$^{-1}$), which is a function of atmospheric stability. $\varepsilon$ is the land surface emissivity, $\sigma$ is the Stefan-Boltzmann constant (5.67 × 10$^{-8}$Wm$^{-2}$ K$^{-4}$), $T_a$ air temperature (°C), $e_p$ is the saturation vapor pressure (kPa) at $T_p$, and $e_a$ is the actual vapor pressure (kPa) at $T_a$. $T_p$ can be obtained through iterations from equations (1) and (2).

For Wet Environment Areal PET (WET), Morton (1983) also modified equation of Priestley and Taylor (1972) to account for the equilibrium temperature dependence of both the available energy and the slope of the saturation vapor pressure curve, i.e.,

$$WET = \frac{1}{\lambda}\left[b_1 + b_2 \frac{\Delta_p}{\Delta_p + \gamma}(R_n - G)_p\right] \tag{3}$$

where $\Delta_p$ (kPa °C$^{-1}$) is the slope of the saturation vapor pressure curve at $T_p$, $(R_n\text{-}G)_p$ is the available energy at $T_p$ (Wm$^{-2}$), i.e., $(R_n\text{-}G)_p = (R_n\text{-}G) - 4\varepsilon\sigma(T_p + 273.15)^3(T_p - T_a)$. $b_1$ accounts for possible advection of energy, significant only during seasons of low net radiation, while $b_2$ is another parameter. The default values of $b_1$ and $b_2$ are given by Morton (1983) as 14 Wm$^{-2}$ and 1.2, respectively and are used in this work.

Morton's areal AET can then be calculated using the complementary relationship:

$$AET = 2WET - PET \tag{4}$$

### 2.4 Observation-based AET datasets

We evaluated seven gridded publicly available observation-based AET products (see Table 2 for more information), which include three Australian AET datasets and four global AET datasets. The performance of these observational-based AET datasets was compared with the high-resolution QldFCP-2 dataset.

**Table 2: List of gridded AET datasets evaluated in this work. RES stands for resolution.**

| Short name | Full name | Res | Time step | Method | Year range | Reference |
|---|---|---|---|---|---|---|



| CMRSET | CSIRO MODIS ReScaled EvapoTranspiration | 30m | Monthly | Satellite-derived | 2000 - current | Guerschman et al., 2022 |
|---|---|---|---|---|---|---|
| AWO | Australian Water Outlook | 5km | Daily | AWRA-L water balance model | 1911 - current | Wilson et al., 2022 |
| SILO | Australian Climate Data | 5km | Daily | Station based interpolation | 1880 - current | Jeffrey et al., 2001 |
| ERA5-Land | Fifth Generation of European ReAnalysis - Land Component | 9km | Hourly | Reanalysis | 1950 - current | Muñoz-Sabater et al., 2021 |
| Copernicuss AET | Copernicus Climate Change Service | 10km | Decadal | Satellite remote sensing | 2000-2018 | Wit et al., 2022 |
| GLEAM | Global Land Evaporation Amsterdam Model | 25km | Daily | Satellite and reanalysis | 1980 - 2022 | Martens et al., 2017; Miralles et al., 2011 |
| DOLCE | Derived Optimal Linear Combination Evapotranspiration | 25km | Daily | Hybrid | 1981-2018 | Hobeichi et al., 2018 |
| QldFCP-2 | Conformal Cubic Atmospheric Model | 10km | Daily | Dynamically downscaling | 1981 - 2100 | This work |

The three Australian observation-based AET datasets include:

a) CMRSET AET (available from TERN) provides high-resolution (30 m) monthly AET for Australia using the CMRSET (CSIRO MODIS ReScaled EvapoTranspiration) algorithm (Guerschman et al., 2022). CMRSET uses satellite-derived indices to estimate AET from PET. The CMRSET AET model was calibrated using the OzFlux dataset, so the output is high-resolution (30m), continuous (no gaps due
to cloud) and accurate.

b) Australian Water Outlook (AWO) provides Australia-wide information on key landscape water balance components including soil moisture, runoff, AET and precipitation using AWRA-L water balance model. The AWO historical sub-collection AET datasets were selected for the purpose of this research.

c) SILO interpolates data from Australian Bureau of Meteorology (BoM) weather stations and additional observational sources to provide daily gridded surfaces for Australia. The gridded SILO dataset covers the period from 1880s to the present and are updated daily. SILO provides AET using Morton's model.

We also compare the QldFCP-2 dataset with four global AET datasets: AET Indicators from Copernicus
Climate Change Service (Copernicus AET, derived from satellite observations), Global Land Evaporation Amsterdam Model (GLEAM, based on satellite and reanalysis data), Derived Optimal Linear Combination AET (DOLCE), and ERA5-Land reanalysis datasets. DOLCE is observationally constrained hybrid AET dataset, derived by merging several global AET datasets. These global datasets were subset to the Australian region and re-gridded to a consistent 10km grid for comparison.

**2.5 Evaluation of AET products**

AET from the observation-based products and QldFCP-2 were evaluated against OzFlux tower data. Evaluations against OzFlux locations considered the nearest grid cell to its location. We calculated bias, absolute error and percentage error for each AET product at OzFlux tower location. Percentage error was calculated as follows:

$$PE = \left|\frac{T-E}{T}\right| * 100 \qquad (5)$$





Here, T = true value (OzFlux towers) and E = estimated value (model or derived from observation).

Note that there are substantial missing values for Copernicus AET datasets, which are excluded from further analysis. In the performance analysis of CMRSET product, we have directly downloaded the AET time series data for each OzFlux tower site aggregated across time at their original high resolution
of 30 m (TERN, 2024) (Arcturus site AET data is not available for CMRSET). For other products, when AET time series data are unavailable for OzFlux towers sites, the nearest neighbour interpolation was used to estimate AET.

### 2.6 Bias correction of AET from downscaled climate simulations (QldFCP-2)

We used quantile delta mapping (QDM) (Cannon et al., 2015) to bias correct derived AET from
downscaled climate simulations, with 1981 – 2020 used as a training period. We bias correct the historical period (1981 – 2014) and the future period (2015 – 2100), for the three emission scenarios to examine the impact of bias correction on mean climatology and climate change patterns. In our QDM implementation, cumulative distribution functions (CDFs) are calculated for each month for individual models and reference datasets. After bias-correcting downscaled simulations, we then
compare the results to the OzFlux dataset.

### 2.7 Evaluation of climate change impacts

We evaluate climate change impacts on future AET and PET in Australia, followed by regional climate change impact analysis. Eight NRM cluster regions are considered, which are grouped together by the super cluster they belong to: a) Northern Australia - Monsoonal North (MN) and Wet Tropics (WT); b)
Eastern Australia - Central Slopes (CS) and East Coast (EC); c) Rangelands (R); d) Southern Australia - Murray Basin (MB), Southern Slopes (SS) and Southern and South-Western Flatlands (SSWF) (refer to Figure 1 for details). The impact of the bias correction on the climate change patterns is evaluated in Australia through comparisons between unaltered projections and bias corrected projections for derived AET and PET.

### 2.8 Controlling factors for AET and PET change

We use a random forest (Breiman, 2001) to identify the major drivers for evapotranspiration changes in regional Australia. Random forest ranks predictor variables by importance and is one of the most-used machine learning algorithms, due to its simplicity and diversity. Random forests models have been utilized in various fields such as in hydrology and in ecology modelling (Trancoso et al., 2016b,
a). While there are many factors which influence evapotranspiration, we focus on the five key climatic variables (precipitation, solar radiation, maximum and minimum temperature, and wind speed) and one vegetation variable (leaf area index - LAI) in our analysis, which was subject to the availability of the output from the downscaled projections. Note that LAI and land surface characteristics do not differ between models. We then assessed the importance of these six variables with random forests
for the eight NMR regions and three seasons (ANN, DJF, JJA). Random forests are a combination of classification trees that randomly select a set of possible explanatory variables without overfit. They are used for both predictions and to assess variable importance. The importance was computed by permutation, where the difference between prediction errors (mean square error - MSE) before and after the permutation of variables was averaged and normalized by the standard deviation. This
procedure considered 3,000 trees. The importance is therefore given by the precent increment of MSE (%IncMSE) when the variable is removed and is used to rank predictor variables (Liaw and Wiener, 2002). In this work, the predictor variables are vegetation and climatic variables, and the response variables are AET and PET climate change signals.

### 3 Results



In what follows, mean annual and seasonal AETs from different products are presented first for validation during the historical period, followed by climate change signals and patterns from the regional climate projections (QldFCP-2).

**3.1 Comparison between QldFCP-2 AET and observation-based AET products**

Figure 2 shows the AET estimates over Australia from the observation-based approaches and the
downscaled projections. While all AET products show similar spatial patterns, regional differences can be seen across all products. For instance, SILO shows larger AET in the north and eastern coastal regions than other products and is generally simpler as it uses limited weather stations. The variability in AET estimates from the different products is the greatest in the central and southern regions. Differences in lake evaporation are evident and captured in ERA5-Land, DOLCE, GLEAM and CMRSET,
but are not seen in SILO, AWO, Copernicus AET and QldFCP-2. In QldFCP-2 lake evaporation was treated separately as an individual evaporation variable. Seasonally, AET is much lower in Austral winter (JJA) and much higher in Austral summer (DJF), particularly along coastal regions. There are substantial grid cells with missing values in Copernicus AET, which might affect its usability in some areas. For the AWO dataset, there are holes (in white colour) for northcentral western Australia,
where precipitation gauges are sparse, resulting in high uncertainty. Thus, for the historical product the AWO model assumes this area has zero rainfall, which results in a zero value for all water outlook components and the appearance of no data in the map. Therefore, caution should be taken for estimated AWO AET across these areas. CMRSET AET provides very high resolution (30m) dataset, which shows finer granularity compared with other coarser AET products.



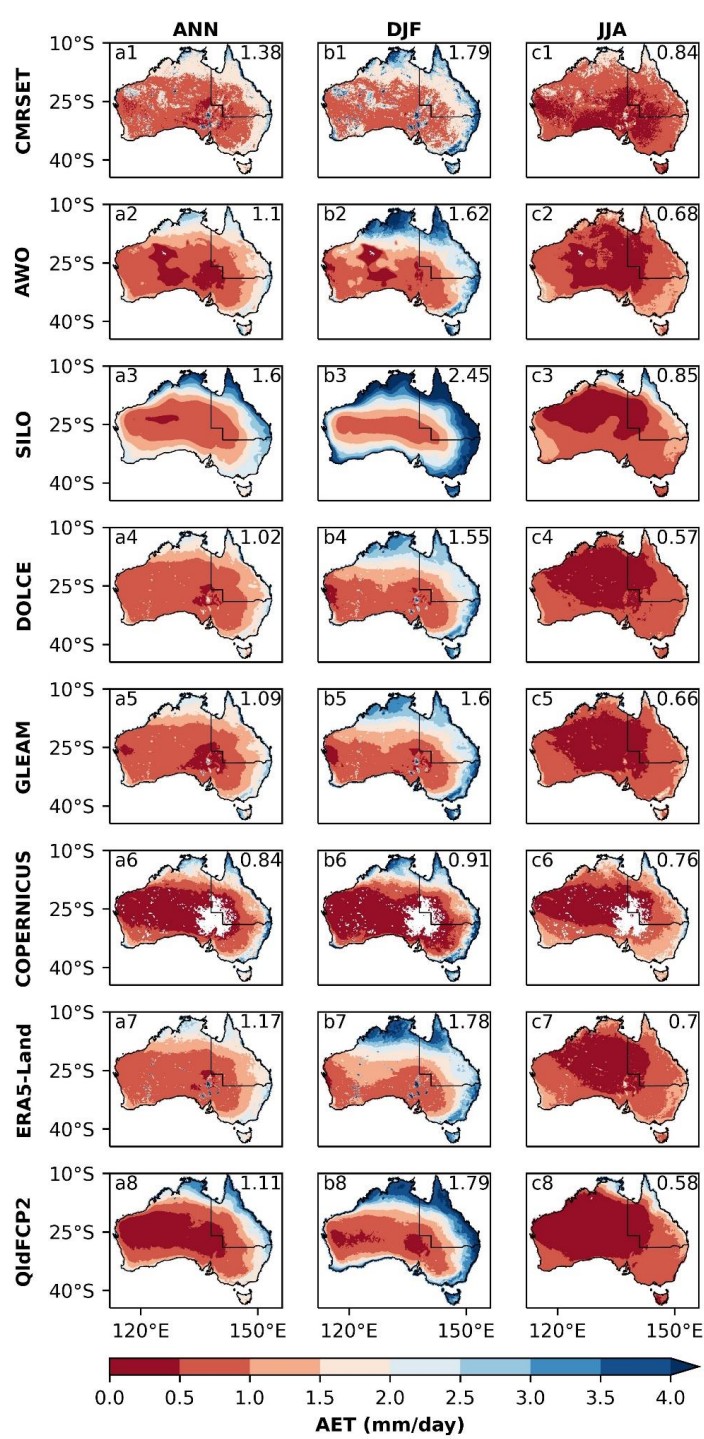




**Figure 2: Annual (column a), Austral summer (December – February; column b), Austral winter (June – August; columns c) of AET for eight datasets, which include four global observation-based or reanalysis datasets (DOLCE, GLEAM, Copernicus AET and ERA5-Land), three Australian observation-based datasets (CMRSET, AWO, SILO) and QldFCP-2 dataset (11-model ensemble average). The period is 30 years (1981-2010) except for Copernicus AET and CMRSET (2000-2018) due to lack of data before 2000. Note that the substantial missing values for Copernicus AET datasets are shown in white colour (row 6). Average across Australian continent shown in top right.**

Of the global datasets, similar spatial patterns are seen for GLEAM, DOLCE, ERA5-Land, while there are some differences with Copernicus AET. Copernicus AET is relatively low in the inland areas, particular in DJF and ANN seasons. Overall, the QldFCP-2 dataset is broadly comparable to the observational AET products.

The results indicate there are still uncertainties in estimating AET, with the largest uncertainties in dry and data sparse regions. AET products in these areas should be used with caution given the observational uncertainties related to Major Australian Desert areas. These regions have been described as a 'data desert' due to their severe aridity and a lack of population.

### 3.2 Point scale AET validation

The performance of the various AET products was evaluated against 26 OzFlux towers in Australia as shown in Figure 3. The percentage errors vary depending on the location of the flux tower. CMRSET AET estimates had the highest agreement with OzFlux and the smallest mean percent errors, averaging 15.7% across all 26 available flux towers (see the last row in Figure 3 for "Mean percentage error"). The ensemble average of the unadjusted QldFCP-2 dataset also performed well, with the second lowest average percent errors (17%) from all flux tower sites. In comparison, SILO had the highest mean percent error at 44%. For other AET products, the mean percent error ranged from 21.5% to 27%. This indicates that QldFCP-2 can provide valuable AET estimates, with similar or less bias to other remote sensing or reanalysis AET products. Within the QldFCP-2 dataset, the percentage errors vary from site to site and across the individual models, with the highest error (110.5%) obtained at the Samford OzFlux tower from the MPI-ESM1-2-LR model and the lowest error (0.1%) obtained at the Warra OzFlux tower from the CNRM-CM6-1-HR atmosphere only model.



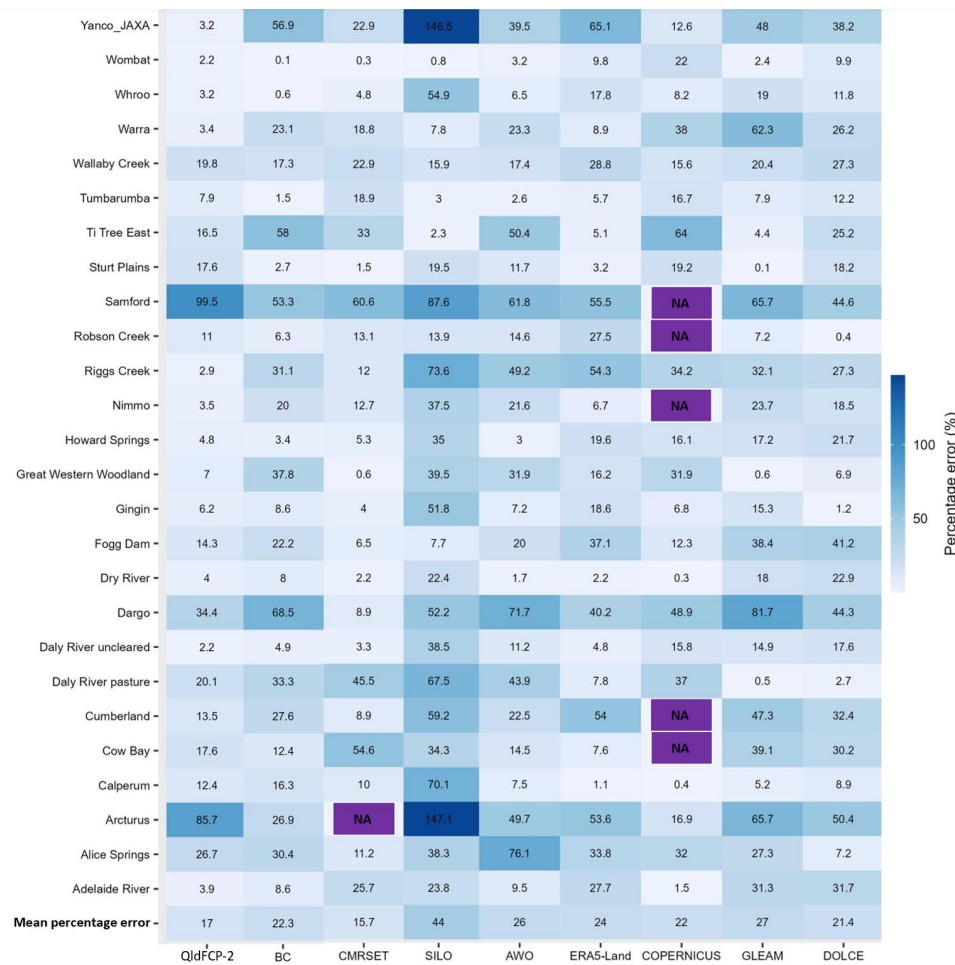

**Figure 3: Percentage errors (%) against 26 site-averaged AET OzFlux tower data for eight comparative AET products, which include four global observation-based or reanalysis datasets (ERA5-Land, COPERNICUS, GLEAM and DOLCE), three Australian observation-based datasets (CMRSET, SILO and AWO), unadjusted QldFCP-2 (11 model average) and bias corrected (BC) QldFCP-2 datasets. Missing values from COPERNICUS AET and CMRSET datasets are expressed as NA.**

The largest mean percentage errors were found at two flux tower sites, Samford and Arcturus for the unadjusted QldFCP-2 datasets. However, other AET products also show relatively high percentage errors at these sites (e.g., SILO product). Bias correction using AWO as reference data can reduce the mean percentage errors for these two sites. However, for other sites, bias correction appears to increase the percentage errors (see the second column in Figure 3). Bias correction of the QldFCP-2 datasets using gridded observation-based AET dataset (e.g., AWO) does not necessarily improve the results. Instead, it makes the bias corrected values closer to the reference dataset, which for the case of CCAM bias corrected with AWO leads to a worse overall mean percentage error of 22.3% compared to 17% for the unadjusted CCAM model (Figure 3). Note that based on the evaluation of the observation-based AET products, we used the AWO AET estimates to bias correct the downscaled projections. The evaluation of the AET products showed CMRSET performed the best, however it only provides monthly data, while daily data are required for bias correction. The reason to include CMRSET





in the evaluation is because it is the latest AET product in Australia and may be useful to users who require monthly, rather than daily data. AWO was selected as the reference dataset as it provides the required daily data for both AET and PET and was the best performing product after CMRSET.

More details for our error analysis are provided in Table S1 and Table S2 in the Supporting Information. CMRSET AET has the best performance in terms of mean percentage errors, which indicates that it best captures the temporal and spatial pattern. The worst performing AET dataset is SILO (column 4
in Figure 3), which uses BoM weather stations to derive AET through interpolation. All AET products have nontrivial errors, with the smallest error being 15.7% for CMRSET. All other AET products have a coarse spatial resolution when compared with CMRSET.

**3.3 Climate change patterns**

Projected AET changes at the end of the century are seen to be scenario, season, and location
dependent (Figure 4 rows 1-3). There are large increases in AET only in DJF for SSP370. Whereas there are decreases in JJA for the dry inland areas, but along the coastal areas and in the northern regions there are still increases due to more available moisture in these regions. In contrast to SSP370, SSP126 and SSP245 projected decreases to AET annually and in summer (or very small positive changes), whilst in winter the decreasing changes are consistent across scenarios. The annual average change
at the end of this century in Australia are 1.8% for SSP370, -3.5% for SSP245 and -4.5% for SSP126.





**Figure 4: Climate change patterns for AET and PET across annual (ANN), summer (DJF), winter (JJA)
seasons in Australia. Projected climate change patterns are shown for 2080-2099 with respect to
1995-2014. Results are shown for 11-model ensemble mean under emissions scenarios (SSP126,
SSP245 and SSP370). Average across Australian continent shown in top right.**

The projected changes for PET behave quite differently from AET (Figure 4 rows 4-6). PET increases in
all three scenarios and across all of Australia, with larger increases in JJA than DJF. The percentage
increases across Australia rise with emissions and radiative forcing. The largest increase in PET occurs
under SSP370 in winter within the southeastern and southwestern regions of Australia. The annual
average increases by the end of this century in Australia are 5.0% for SSP126, 8.4% for SSP245 and
11.5% for SSP370.

Bias correction impacts the climate change patterns (Figure S1), with greater impacts for AET than
PET. For PET, bias correction only changes the climate change patterns slightly, with reduced annual
or seasonal average increases in Australia except for SSP126 in JJA. Bias correction increases annual
and seasonal AET for most seasons and scenarios except for SSP126 in JJA. Note that the 'holes' in the
bias corrected maps for AET are artefacts, denoting the limitation of reference AWO in data sparse
regions. This affects the climate change patterns and illustrates the importance of quality
observational datasets to enhance bias correction performance.

**3.4 Regional climate change impacts**





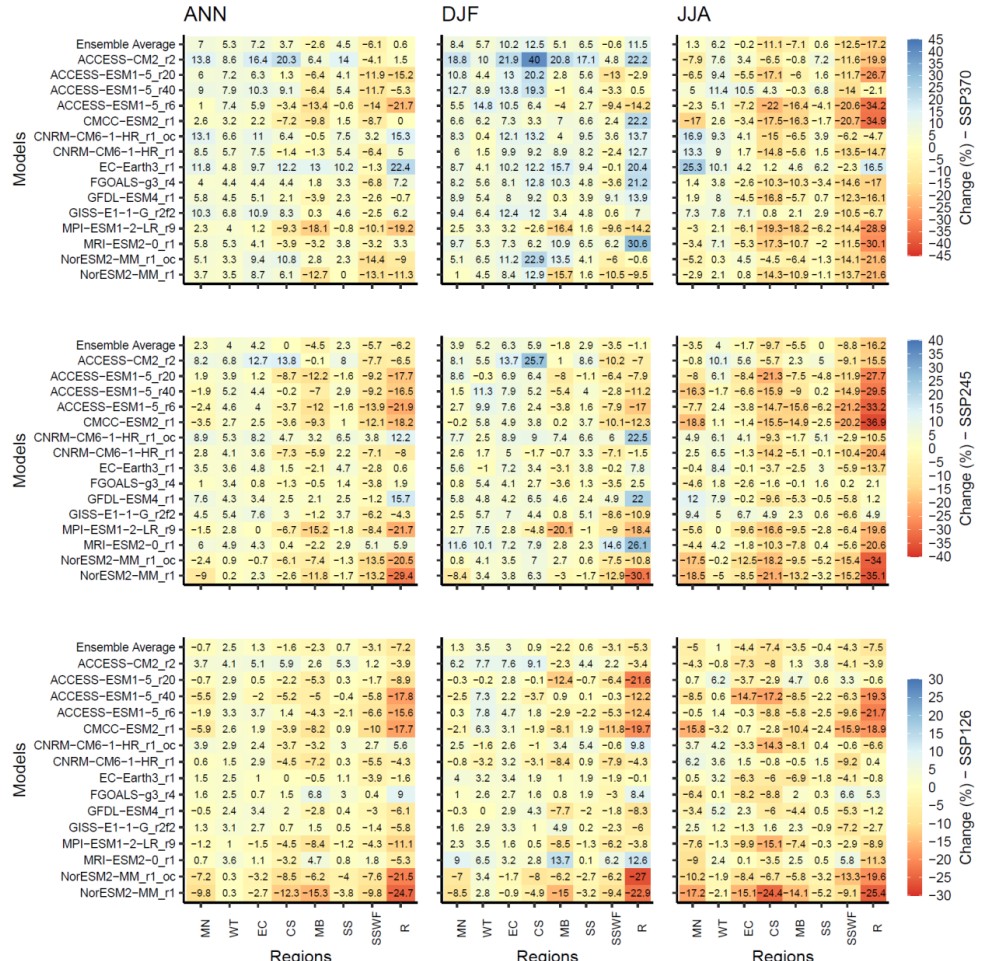

**Figure 5: Heatmaps for climate change signal (percentage difference between future and present) from 1995-2014 to 2080-2099 for AET across annual (ANN), summer (DJF), winter (JJA) seasons in 8 NRM regions in Australia. Projected changes from QldFCP-2 under emissions scenarios (SSP126, SSP245 and SSP370) are shown for individual models and ensemble mean in each row.**

Next, we show the climate change impact on AET over eight NRM regions (Figure 5).

**RANGELANDS (R):** This is one of the driest regions. In JJA, most downscaled projections show the greatest decreases across all emission scenarios. In DJF, models generally project increases to the climate change signal with increasing emissions. In DJF, the SSP370 is the only scenario where consistent increases in AET are projected. Annually, for low and medium emission scenarios (SSP126 and SSP245), models generally project a decreasing signal, whereas for SSP370, increases are projected by the majority of models. On average, the annual change by the end of the century is negative for the low and medium emission scenarios (SSP126 and SSP245), and slightly positive under high emissions (SSP370).

**Eastern Australia (CS and EC):** In JJA, most models project moderate decreases for all three emission scenarios, whilst in DJF, most models project increases, particularly for the moderate and high





emissions scenarios (SSP245 and SSP370). On average, the annual change at the end of this century is small for the low and medium emission scenarios (SSP126 and SSP245), but it is increased under high emissions (SSP370).


**Southern Australia (SSWF, MB, and SS)**: In SSWF, most models project moderate decreases regardless of seasons and emission scenarios. In MB, most models project moderate decreases except for high emissions (SSP370) in DJF. In SS, most models project moderate increases except for low emissions (SSP126) in JJA.

**Northern Australia (WT and MN)**: In WT, most models project increases regardless of seasons and emission scenarios. As emissions rise, the increases become larger. In MN, most models project moderate increases in DJF. In JJA, models project increases with greater emissions. Annually most models project moderate increases except for low emission (SSP126).

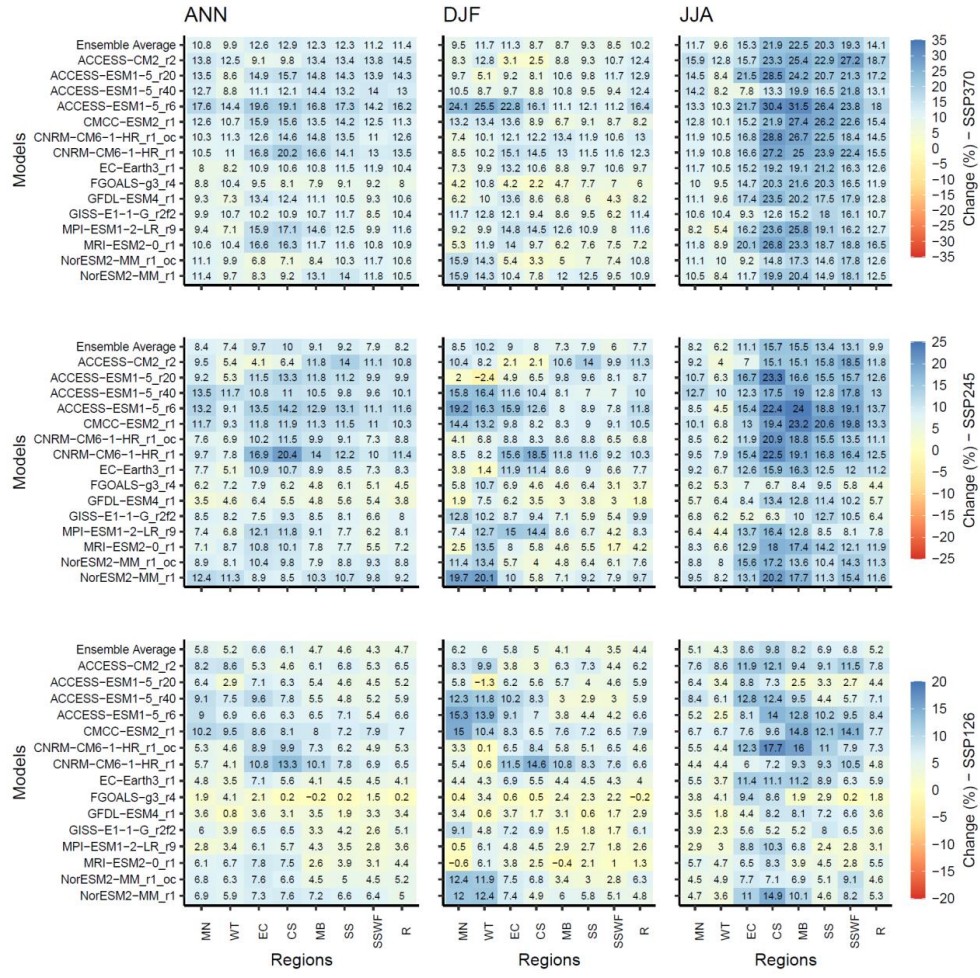


**Figure 6: Percentage climate change signal (2080-2099 relative to 1995-2014) in the eight NRM regions for individual models and the model average in each row for PET.**





For PET, generally most model projections show increasing changes (Figure 6), with greater increases as the emission scenario increases across all regions, models, and seasons. Generally, there are larger increases in JJA compared to DJF, particularly for the CS, MB, EC, SS and SSWF regions. In comparison, MN and WT regions have relatively small increases in JJA. In DJF, the WT region has the largest increase followed by the EC and MN regions, while the SSWF region has the smallest increase. Annually, CS and EC have the largest increases whereas WT and SSWF have the smallest increases in model averages across the three scenarios.

Though there are variations from model to model, the spread of the seasonal climate change signal is smaller for PET (Figure 6) compared to AET (Figure 5), indicating smaller uncertainties for PET projections. Thus, there is high likelihood that there will be an increase in PET by the end of this century regardless of model, scenario, or season.

**3.5 Drivers for climate change impacts on AET and PET**

Precipitation and solar radiation are shown to be the two most important drivers for AET change for most regions across Australia (Figure 7). For example, in the Rangelands (R) region under SSP245, the annual average % increase in mean square error (MSE) is the highest for precipitation (62.69), followed by solar radiation (60.94), and wind speed (31.17). For R region and other two scenarios (SSP126 and SSP370), precipitation or solar radiation also ranked the top two most important drivers, followed by wind speed. For other regions such as MB, SSWF, CS, EC, the % increase in MSE (%incMSE) for precipitation and radiation shows decreasing value, but still maintain the top two most important drivers for AET change. Seasonally, the %incMSE for each predictor variable varies. The highest %incMSEs appear in DJF for precipitation and solar radiation in R region, which reaches 75.62 for precipitation and 66.39 for radiation for emission scenario (SSP245). In DJF, precipitation, solar radiation and wind are the dominant drivers for regions such as R, MB, and SSWF. Maximum or minimum temperature also plays a role for other regions such as WT, MN, EC, CS and SS. In JJA precipitation and solar radiation are the most important drivers for most regions such as R, MN, MB, CS and SSWF except for WT and SS regions where maximum or minimum temperature plays a significant role. Minimum temperature has the highest scores for several seasons and scenarios in humid and saturated region like WT and SS so atmospheric demand is more important than supply.





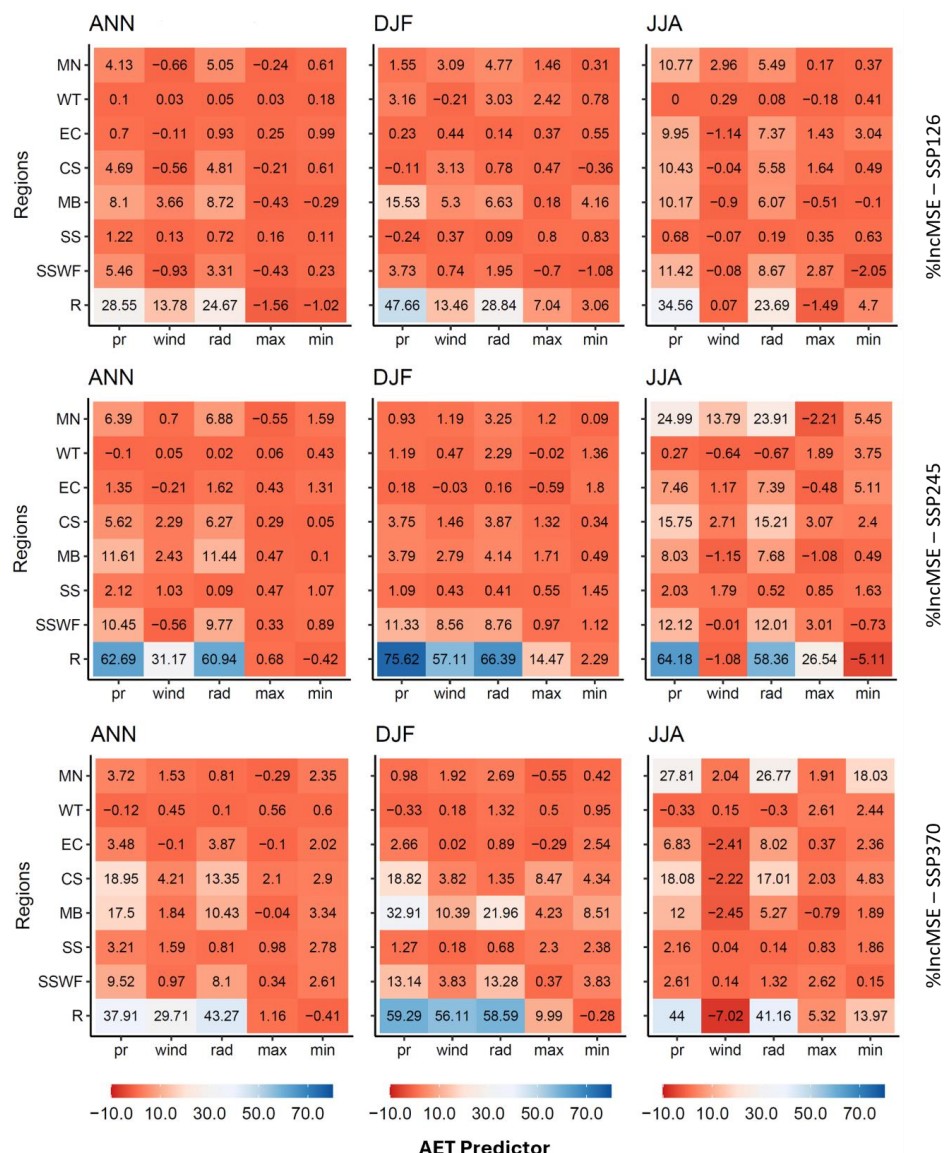

**Figure 7: Heatmaps showing the importance (%IncMSE) of AET predictors (precipitation - pr, wind speed - wind, solar radiation - rad, maximum temperature – max and minimum temperature – min) across annual (ANN), summer (DJF), winter (JJA) seasons. The random forest model is applied to eight NMR regions and three scenarios for the climate change signal (from 1995-2014 to 2080-2099). The rows represent eight NRM regions, and columns represent the five predictors.**


As the climate change signal and pattern for AET behave quite differently from those of PET, we repeated the random forest analysis for PET. Solar radiation and maximum temperature are shown to be the most important drivers for PET change in most regions (Figure 8). Seasonally, the highest






%incMSEs appears in DJF for solar radiation and maximum temperature in the WT and MN regions. In DJF, while solar radiation and maximum temperature are the dominant drivers for most regions, precipitation also plays a role in some regions such as in MN, WT, and CS. In JJA solar radiation and maximum temperature are the most important drivers for most regions, followed by precipitation. However, in the SS region, minimum temperature is also important. This demonstrates that the controlling factors for PET change are dominated by solar radiation and max temperature. As air temperature is closely related to solar radiation (Shrestha et al., 2019), solar radiation plays the most important and direct role to drive PET change, followed by maximum temperature (solar radiation heats the surface, which results in increases in the air temperature).


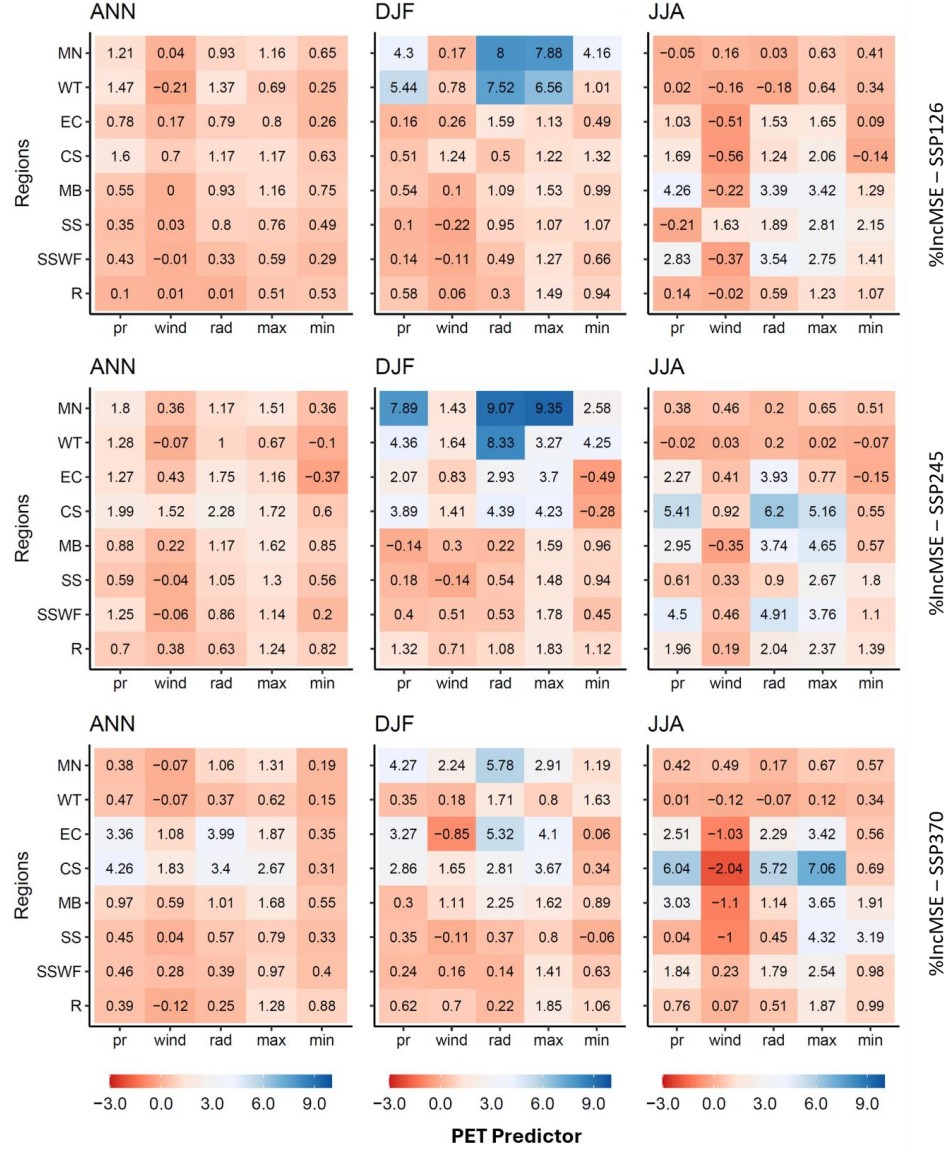


**Figure 8: The same as Figure 7 except for PET.**



Vegetation factor (LAI) is also considered as AET and PET predictor variable and its importance is presented in Figure S2 (together with other five climatic variables). Only one model (EC-Earth3) is considered in Figure S2 as there is no variation in LAI among different CCAM models. Figure S2 (a) shows that LAI plays some important roles for some regions (e.g., SS), but it is not the principal controlling factor for most regions and seasons. Annually precipitation plays the most important role for AET change in the MN and R regions, while solar radiation is the most important factor in WT, SS, and SSWF and minimum temperature is the primary factor in the EC and CS regions. Seasonally precipitation and solar radiation are still the most important drivers over most regions, followed by LAI. From Figure S2 (b), we can see that solar radiation and maximum temperature are the principal factors controlling PET changes for most regions and seasons, followed by LAI, minimum temperature and wind. Annually solar radiation is the most important factor for determining PET change in all regions, followed by maximum temperature. Seasonally, these two factors are still the dominant factors for most regions, with LAI only playing an important role in MN region in JJA.

**4 Discussion**

This study assesses seven observation-based AET products and QldFCP-2 downscaled AET simulations against 26 OzFlux measurements. Based on their accuracy assessment, agreement between eight AET datasets and the flux tower measurements followed the order: CMRSET > QldFCP-2 > DOLCE > Copernicus AET > ERA5-Land > AWO > GLEAM > SILO. The QldFCP-2 AET appears to have comparable spatial patterns of AET to most of the observation-based AET products. The mean percentage error from the range of observation-based AET estimates were between 15.7% to 44%, indicating substantial uncertainty in these products. This highlights the difficulty in accurately estimating AET using satellite-based remote sensing and global inversion products and shows the need for continual ground-truth monitoring through flux towers in order to validate and improve these products. SILO AET was the worst performing dataset analysed, which was likely due to limited weather data inputs to derive AET and the lack of calibration against the OzFlux dataset. It is interesting to note that the downscaled QldFCP-2 AET performed better than many of the observational products, and as such bias correcting to these datasets may actually introduce more errors when compared to the unadjusted dataset. These results highlight the importance of evaluating gridded observation datasets against high quality point data, such as flux towers, prior to adopting them for bias correction.

The performances of the different observational AET products varied substantially across the 26 OzFlux AET measurement sites. For the Samford and Arcturus sites, the disagreements between models and OzFlux AET measurements were relatively large. The disagreements here may be due to model performance, or measurement issues at these sites. The Samford flux station is located on improved pasture, approximately 20 km west of the Brisbane in south-eastern Queensland (the dashed line in Figure 1 shows the Queensland border). Measured mean annual AET is 564.2 mm/year, while measured mean annual rainfall is 1240.6 mm/year for this site. For Samford, AET models overestimate AET with percentage errors ranging from 44.65% for DOLCE to 99.53% for QLDFCP-2. The Arcturus monitoring station is located 48 km southeast of Emerald, Queensland, with measured mean annual AET of 398.9 mm/year and measured mean annual rainfall of 638.8mm/year. For the Arcturus site, models also generally overestimate AET with percentage errors ranging from 16.89% for Copernicus to 147.05% for SILO. Modelled AET values are even larger than mean annual precipitation measurements for some AET products. Such overestimations might be due to the modelled AET datasets not being able to represent topography or land cover adequately over these sites.

We evaluated the impact of climate change on AET and PET across Australia. For SSP125 and 245, annual AET was projected to decrease by the end of this century. For SSP370, annual AET was projected to increase slightly, while in summer (DJF) most models projected increased AET. On the other hand, in winter (JJA) most models projected decreased AET. Our results generally point towards a pattern of greater increases in AET for the higher emissions scenarios when compared to the lower scenarios. In contrast, PET was projected to increase across all scenarios and seasons. These findings



for PET using downscaled CMIP6 models are broadly in agreement with previous results for PET increases using CMIP5 models in southeastern Australia based on statistical downscaling (Shi et al., 2020) and for the continental Australia based on four global climate models and CCAM (Peter et al., 2023). The spatial patterns of increasing in PET are generally consistent with the two studies above

assessing CMIP5 models, but studies to date have not focussed on AET changes in Australia, have not used the latest CMIP6 projections, nor the associated impacts for populated regions.

The analysis above highlights the climate change impact over various NRM regions in Australia. Among them, the Rangelands region dominates Australia in terms of area size (about 80% of Australia). Changes in AET over agricultural areas in NMR regions are important as it impacts on crop production.

Similarly, changes in AET over urban water supply catchments also have big impacts in water security. Meanwhile, in terms of population, more than 85 per cent of Australians live within 50 kilometres of the coast, with the population in the eight capital cities accounts for roughly half of the Australian population (the blue circles in Figure 1 shows the location of the eight capital cities).

Our results present a significant contribution towards advancing our understanding on the expected

signs of changes for AET under a range of low to high emissions scenarios. The regions subject to increased future AET include WT, MN, SS (including Tasmania), Eastern and Northern coastal areas. The latter coastal areas cover a vast fraction of Australian population, who would be affected by substantially increased AET by the end of this century. As emissions rise from intermediate to high levels, AET increases generally intensify. This will have large impacts on society, as many of these

regions are already facing water scarcity and have experienced severe droughts in the recent past (Steffen et al., 2018). Consequently, highly populated regions across Australia need to prepare for larger evaporative losses, including accounting for these projected future conditions in adaptation policies. Although the Rangelands region dominates Australia in terms of area size, most of the Australian population, is concentrated along the eastern seaboard. Therefore, the impact of AET

change over Australia is disproportional, more strongly affecting the major capital cities in the eastern coastal areas. There were, however, some regions where AET was projected to decrease including in SSWF, R and MB, particularly in JJA and under low to intermediate emissions. This may be due to very limited precipitation and moisture in the dry inland areas in Australia (or may be due to the limitation of future land cover projections, which has some influences on these results).

Overall, our individual models yielded generally consistent increases for annual PET and across all calendar seasons. However, for AET individual models generally had lower agreement. This is likely to be linked to the higher climate sensitivity of CMIP6 models to precipitation (Zelinka et al., 2020). Maximum temperature is projected to increase in all seasons and scenarios with climate change (Chapman et al., 2024), whereas the climate change signal for precipitation is highly uncertain and

varies between models (Chapman et al., 2024), which explains the heterogeneity of the AET climate change signal. The PET signal however, which is not dependent on precipitation, shows clear increases across Australia. This is in line with previous work, which have also shown that precipitation is the primary driver of AET in Australia (Li et al., 2022; Liu et al., 2022b). Li et al. (2022) found that the precipitation variability is the principal control for global AET in dry climates (precipitation controls

AET changes across almost the whole of Australia), while the net radiation has substantial control over AET in tropical regions, and vapor pressure deficit (VPD) impacts AET changes in the boreal mid-latitude climate. Meteorological factors and vegetation activity can have different effects on AET at different time scales and in different regions (Jiang and Liu, 2022; Liu et al., 2018, 2019; Yang et al., 2022). For example, Yang et al. (2022) found that precipitation, temperature, and normalized

difference vegetation index (NDVI) were the most important factors controlling AET from 1982 to 2015 in northwest China.

The climate change signals for both AET and precipitation are shown in Figure S3 for comparison. Figure S3 indicates that the climate change signal for AET is very similar to that for precipitation for most individual models, seasons, and scenarios. This indicates that there will be less winter



precipitation in the future, which contributes to a decrease for AET in winter due to a lack of moisture to allow AET to occur (though PET is still increasing). On the other hand, in DJF, more consistent increases to precipitation emerge across Australia as emission scenario increases from SSP126 to SSP370, with AET typically following suit. This indicates that there will be more summer precipitation in Australia under SSP370, allowing evapotranspiration to occur in this season due to additional
moisture, resulting in an increase for AET. Annually the increasing changes and the decreasing changes for the different seasons and models even out, resulting in a minor increasing change for both precipitation and AET under SSP370.

For PET, the availability of water (from precipitation and soil moisture) is unlimited and so precipitation has little influence, with solar radiation instead found as the principal controlling factor
explaining future changes. Guo et al. (2017) assessed the factors that have an impact on PET at 30 Australian locations representing different climatic zones and found that temperature was the most important variable for PET. In other regions such as Northeastern Asia, studies from historical data analysis showed that PET was predominantly influenced by temperature and VPD (You et al., 2019). Based on a multi-source global remote sensing datasets, Liu et al. (2022) showed that temperature
was the primary control of PET in more than 85% of the world.

Using high resolution CIMP6 downscaled climate models, we can provide reasonably accurate estimations for AET in Australia (i.e., through historical evaluations), and project AET and PET changes at the end of this century. However, it is important to note several uncertainties associated with this work. Firstly, AET and PET are complex phenomena and may not be represented ideally using one
model (e.g., Morton's model). Thus, comparison with other nonlinear-CR models or water balance formulas may provide further insights in future studies. Second, the uncertainty surrounding emissions scenarios and the CMIP6 downscaled projections leads to subsequent uncertainty for future projections of AET in Australia and in the eight NMR regions. Lastly and most importantly, the uncertainty in AET projections is closely related to precipitation uncertainties of CMIP6 projections,
which vary significantly across Australia (Trancoso et al., 2024). Despite these limitations, our study provides valuable outcomes, which will be useful for policymakers and scientists to establish climate change adaptation and mitigation strategies and to effectively manage water resources.

**5 Conclusions**

In this work we evaluated dynamically downscaled climate simulations and gridded observation-based
AET data against site-specific AET data from OzFlux towers. We find that the dynamically downscaled QldFCP-2 datasets are comparable with observation-based AET datasets in Australia, with comparable errors (17% for the QldFCP-2 ensemble average, which is larger than 15.7% for CMRSET AET product, but smaller than most other models: 44% for SILO, 26% for AWO, 24% for ERA5-Land, 22% for Copernicus AET, 27% for GLEAM and 21.45% for DOLCE). Given the good skill of QldFCP-2 derived AET,
bias correction does not necessarily improve the agreement with OzFlux data.

We also projected PET and AET changes at the end of this century in Australia using new downscaled CMIP6 datasets as input. We found that the climate change signal for AET is scenario dependent in Australia, with decreases projected for low and medium emission scenarios (SSP126 and SSP245) and minor increases under high emission scenario (SSP370) by the end of the century (2080-2099 relative
to 1995-2014). PET will increase under all three scenarios.

From random forests analysis, we found that the main drivers of AET and PET change cannot be generalized from region to region. However, some important drivers such as radiation and precipitation explain the variability of AET characteristics at most regions with differing importance. Our findings also suggest that climatic and vegetation properties have a significant effect on AET and
PET changes at regional scales. However, the relative importance of these properties varies among regions.



Importantly, for larger countries such as Australia, spatial means across the whole country may be too generalised to inform policies. Therefore, regional projections as described in this work are more suitable to inform decision making.


**AUTHOR CONTRIBUTIONS**

**Hong Zhang:** Conceptualization, Methodology, Software, Writing - Original draft preparation.

**Sarah Chapman**: Figure preparations, Data visualization and curation, Writing – draft preparation, review and editing.

**Ralph Trancoso**: Visualization, Investigation, Methodology, Random forest suggestion, Supervision, Writing - Reviewing and editing, Funding acquisition.

**Nathan Toombs**: Software, Validation.

**Jozef Syktus:** Conceptualization, Downscaling simulations data, Methodology, Funding acquisition, Writing - Reviewing and Editing.

**Rohan Eccles:** Conceptualization, Data validation and test, Visualization, Writing - Reviewing and Editing.

**CONFLICT OF INTEREST**

The authors declare no conflicts of interest.

**FUNDING INFORMATION**

This research was supported by the Queensland Government, Department of Energy and Climate and Department of Environment and Science.

**DATA AVAILABILITY**

The downscaled QldFCP-2 data used in this study is available via the CORDEX Australasia domain archive. The 20km data for the Australasian CORDEX domain is available from NCI (National Computational Infrastructure): https://dx.doi.org/10.25914/8fve-1910 and 10 km resolution data is available from NCI: https://dx.doi.org/10.25914/2c0z-8t40. ET data are available via request.

Data from QldFCP-1 simulations and ET data are available from long paddock portals and TERN https://www.longpaddock.qld.gov.au/qld-future-climate/data-info/tern/ (Syktus et al., 2020a) and https://portal.tern.org.au/metadata/TERN/bf437edd-a533-4967-ad46-b1cb1dc3ac82 (Syktus et al., 660 2020b).

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
