# Peer review of "Projections of actual and potential evapotranspiration from downscaled high-resolution CMIP6 climate simulations in Australia"

_EGUsphere, 2025_

## Author Comment (AC1)

**Response to Reviewer's comments for manuscript entitled "Projections of actual and potential evapotranspiration from downscaled high-resolution CMIP6 climate simulations in Australia" [MS No. egusphere-2025-498] submitted to HESS**

| ID | REVIEWER COMMENT | RESPONSE | STATUS |
|---|---|---|---|
| *Reviewer #1* | | | |
| R1.1 | The manuscript presents a very thorough and broad assessment of AET and PET datasets and projections for Australia. The introduction, analysis and discussion give a good overview of the performance of the available AET and PET products. The paper is well written, clearly structured and I appreciate that the limits of the products and the results are clearly stated. | We appreciate the time you spent providing helpful feedback to improve our manuscript. Below we outline the changes we will make to the manuscript to address your comments. | NA |
| R1.2 | First, the abstract would benefit from more plain language. While abbreviations like AET and PET are defined, CCAM is introduced without explanation. Removing it or adding a brief description of what CCAM refers to and why it is used would make the abstract more accessible to non-specialist readers. | We will modify the abstract to reduce complexity as suggested by the reviewer. This will include removing any reference to "CCAM", which we will term "downscaled CMIP6 models" instead to make it more intelligible to readers. | To be implemented |
| R1.3 | The introduction is very thorough but also very long, and the study's aims only become apparent at the end. The mention of "old observational datasets" seems abrupt and lacks sufficient context. Clarifying how these datasets relate to the objectives would improve readability and make it easier to follow. | We will revise the Introduction to improve logical flow and readability. The study's aims will be introduced earlier (e.g., in the second paragraph in Introduction) as suggested. We will downsize the introduction, by moving some parts to the methods to improve readability and emphasize the objectives of our contribution. | To be implemented |
| R1.4 | The initial visual comparison of datasets across Australia is a helpful starting point, but it would be even more informative if it were supplemented with a quantitative assessment of spread or uncertainty between the datasets, e.g. a map of the model spread. Specifically, identifying regions with the largest disagreement among models | We will implement the signal-to-noise ratio analysis to quantify the model spread and uncertainty of the projected changes of the spatial maps. These changes will be made to Figure 4 and Figure S2. We have attached an example of the proposed changes to be made to Figure 4 below. | To be implemented |

would highlight areas where confidence is lower and further improvement is needed.

[Figure]

We will also add the text below in the methods explaining the approach.

"*We examined the signal-to-noise ratio to determine where the climate change signal emerges from the 'noise' of the model ensemble (Hawkins and Sutton, 2011). Here, we take the signal as the model ensemble average, while the noise is calculated as the standard deviation of all the projections. As we focus on end of century climate change impacts, model uncertainty is*

*considered as noise and is expected to be the greatest source of uncertainty. Stippling is shown on ensemble mean change maps where the signal-to-noise ratio is greater than 1.0 (Chapman et al., 2024), indicating agreements among models and implying a higher degree of confidence."*

We will also include an additional figure (Figure 5) to outline the spread of the projections from all models for all emissions scenarios across Australia by the end of the century for both AET and PET. This will consist of a scatterplot with individual model changes in AET and PET as percent and marginal boxplots to highlight the model spread and uncertainty. We have included the proposed plot and caption below.

[Figure]

*Figure 5. The percentage change in annual AET and PET for the individual downscaled CMIP6-CCAM models across Australia for SSP126, SSP245, and SSP370 (1995–2014 compared to 2080–2099). The box and whisker plot shows the interquartile range (box), and the median (bar), while the whiskers extend from the box to the furthest datapoint within 1.5x the interquartile range. The symbol "+" shows the ensemble average and the symbol "x" indicates the outliers from the marginal boxplots.*

In accordance with the proposed changes to these figures, we will edit the Results to add interpretation of these new results. The proposed text to be added to the Results section is below.

*"For AET, there are a few areas where the signal-to-noise ratio is greater than 1, most notably along coastal eastern and northern Australia. Generally, model agreement is greater in DJF than in JJA, and greater for the high emissions scenario (SSP370) than the moderate or low emissions scenarios. By contrast, PET can be seen to have generally had a widespread model agreement according to the signal-to-noise ratio across the whole country, with a few minor exceptions. These differences relate to the very clear increases noted for PET due to increasing temperatures, which are not reflected in AET due to the majority of Australia being water-limited rather than energy-limited.*
*While there is a clear sign of an increase in PET across Australia by the end of the century for all the models considered across all emissions scenarios (Figure 5), the magnitude of the changes can be seen to vary among individual model members. By contrast, for AET, there is disagreement among the individual model members on the sign of the change. For example, for SSP126 while most models show a decreasing signal, there are two models which project increases. For the moderate emission scenario (SSP245) most models project decreases, whereas for the high emission scenario (SSP370), most models project increases. Even when using the same emissions scenario, the projected changes in AET can differ significantly among models, highlighting a key aspect of climate modelling uncertainty and variability in the projections."*

*And:*
*"After bias correction the model agreements have been improved for AET, particularly for SSP370 in DJF season and ANN (refer to the stippling in Figure S1 rows 1-3). For PET, bias correction also improved the consistency across models in some regions, as the signal-to-noise ratio was noted to be greater than 1 across nearly the whole country (Figure S1 rows 4-6)."*

| | | | |
|---|---|---|---|
| | | | |
| R1.5 | Adjusting the order of SSP scenarios in Figure 6 to the order used in Figures 7 and 8 would make it more intuitive to interpret the results. Also, I would include the caption entirely for Figure 8, even if it is the same as for Figure 7, so that the figure can stand on its own and readers don't have to jump between the two. | We will adjust the order of SSPs in Figures 5 and 6 to the order used in Figures 7 and 8. Also, we will include the caption entirely for Figure 8. | To be implemented |
| R1.6 | Lastly, regarding the drivers of change, CMIP6 models usually include LUC in their scenarios. It would be interesting to discuss if some of the changes you identify can be tied to LUC instead of CC. What do you think? | As suggested, we will include the following sentences in the discussion to explain how these changes in AET and PET are influenced by land use changes, and not just climate change: *"Regarding the drivers of change in AET and PET, some of the changes we identify can be tied to land use change, not just climate change. For example, deforestation or agricultural practices can alter surface water availability and vegetation cover, impacting AET, while changes in land surface properties (like albedo) can affect PET."* | To be implemented |

---

## Author Comment (AC2)

**Response to Reviewer's comments for manuscript entitled "Projections of actual and potential evapotranspiration from downscaled high-resolution CMIP6 climate simulations in Australia" [MS No. egusphere-2025-498] submitted to HESS**

| ID | REVIEWER COMMENT | RESPONSE | STATUS |
|---|---|---|---|
| *Reviewer #2* | | | |
| R2.1 | The manuscript uses dynamically downscaled CMIP6 datasets as input to estimate Actual Evapotranspiration (AET) and Potential Evapotranspiration (PET) under historical and various future climate scenarios. It also employs a random forest approach to identify key driving factors influencing projected changes in AET and PET. In addition, the study evaluates multiple datasets against site observations, providing a valuable reference for selecting appropriate AET or PET products. I appreciate the authors' efforts in conducting these evaluations and projections. However, several issues should be addressed before the manuscript is suitable for publication. | Thank you very much for your review and constructive comments. We appreciate the time you have spent reading, reviewing and writing this report. Below we outline how we plan on addressing each of your comments. | NA |
| R2.2 | 1. Justification of CMIP6 model selection
The authors should clarify the rationale behind the selection of specific CMIP6 models and ensembles. Why were these models chosen? Do other CMIP6 models not provide the relevant variables? A brief explanation would help readers understand the basis for the selection. | We will include additional details in the methods section to justify the CMIP6 model selection:
*"The ensemble of CMIP6 models chosen for downscaling in this work was selected considering the models with best skills representing the Australian historical climate, while capturing the future spread in the climate change signal from the full ensemble of CMIP6 models, and prioritizing independent models (Trancoso et al., 2023). The analysis was based on the Kling-Gupta efficiency (KGE) for temperature, precipitation and sea surface temperature and the future climate change signal. An overall skill score for historical simulations was calculated for every ensemble, which was then used to select the best performing ensemble runs across the future envelope of changing temperature and precipitation."* | To be implemented |

| R2.3 | 2. Consistency of projections among models
The manuscript uses the mean values from multiple CMIP6 projections. However, it is unclear whether the individual models indicate consistent changing trends (e.g., all showing an increase or decrease). Are there any models that suggest an opposite direction of change, which may have been masked by averaging? This should be discussed to provide a clearer picture of the uncertainty and variability in the projections. | To assess the consistency of the projections, we will include an additional figure (Figure 5 below) outlining the spread of the projections from all models for all emissions scenarios across Australia by the end of the century for both AET and PET. We will also update our spatial maps of projected changes to include the signal-to-noise ratio to highlight where the climate change signal emerges from the noise of the ensemble of climate models. In accordance with these changes, we will revise our results section as below.

*"For AET, there are a few areas where the signal-to-noise ratio is greater than 1, most notably along coastal eastern and northern Australia. Generally, model agreement is greater in DJF than in JJA, and greater for the high emissions scenario (SSP370) than the moderate or low emissions scenarios. By contrast, PET can be seen to have generally had a widespread model agreement according to the signal-to-noise ratio across the whole country, with a few minor exceptions. These differences relate to the very clear increases noted for PET due to increasing temperatures, which are not reflected in AET due to the majority of Australia being water-limited rather than energy-limited.*
*While there is a clear sign of an increase in PET across Australia by the end of the century for all the models considered across all emissions scenarios (Figure 5), the magnitude of the changes can be seen to vary among individual model members. By contrast, for AET, there is disagreement among the individual model members on the sign of the change. For example, for SSP126 while most models show a decreasing signal, there are two models which project increases. For the moderate emission scenario (SSP245) most models project decreases, whereas for the high emission scenario (SSP370), most models project increases. Even when using the same emissions scenario, the projected changes in AET can differ significantly among models, highlighting a key aspect of climate modelling uncertainty and variability in the projections."*

And: | To be implemented |

*"After bias correction the model agreements have been improved for AET, particularly for SSP370 in DJF season and ANN (see the more stippling areas in Figure S1 rows 1-3). For PET, bias correction also improved the model consistency in some regions, as the signal-to-noise ratio was noted to be greater than 1 across nearly the whole country (Figure S1 rows 4-6)."*

[Figure]

*Figure 5: The percentage change in annual AET and PET for the individual downscaled CMIP6-CCAM models across Australia for SSP126, SSP245, and SSP370 (1995–2014 compared to 2080–2099). The box and whisker plot shows the interquartile range (box), and the median (bar), while the whiskers extend from the box to the furthest datapoint within 1.5x the interquartile range. The symbol "+" shows the ensemble*

| | | *average and the symbol "x" indicates the outliers from the marginal boxplots.* | |
|---|---|---|---|
| R2.4 | 3. Definition of CCAM
The abstract mentioned CCAM without defining it. The full name should be provided upon first mention. | We will remove reference of "CCAM" in the abstract which we instead term "downscaled CMIP6 models" in order to reduce complexity. We introduce "CCAM" with its full definition in the main text, where we can explain the model without space constraints. | To be implemented |
| R2.5 | 4. Improving logical flow in the Introduction.
The introduction could benefit from improved coherence. While the authors have evaluated both AET and PET, the transitions between topics are sometimes abrupt. For instance, around line 55, the discussion shifts from AET to PET and then back to AET, which disrupts the logical flow. Strengthening the narrative structure would enhance readability. | The Introduction will be revised to improve logical flow and readability. We will add more information about PET (e.g., control factors and PET applications) to improve the transitions between AET and PET. Also, we will restructure the Introduction accordingly to strengthen the narrative structure and enhance readability. | To be implemented |
| R2.6 | 5. Figure 1 caption clarity
The caption for Figure 1 does not explain the meaning of the solid and dashed lines, which makes it difficult to interpret the boundaries of the eight Natural Resource Management (NRM) regions. Although the regions are numbered, a clearer description of the line styles is needed. | We will include an additional sentence in the caption of Figure 1 to explain the meaning of the solid and dashed lines as suggested:
"*The solid lines represent the boundaries of the eight Natural Resource Management (NRM) regions, and dashed lines represent the boundaries for Queensland.*" | To be implemented |

---

## Author Response (AR1)

**Response to Reviewer's comments for manuscript entitled "Projections of actual and potential evapotranspiration from downscaled high-resolution CMIP6 climate simulations in Australia" [MS No. egusphere-2025-498] submitted to HESS**

| ID | REVIEWER COMMENT | RESPONSE | STATUS |
|---|---|---|---|
| *Reviewer #1* | | | |
| R1.1 | The manuscript presents a very thorough and broad assessment of AET and PET datasets and projections for Australia. The introduction, analysis and discussion give a good overview of the performance of the available AET and PET products. The paper is well written, clearly structured and I appreciate that the limits of the products and the results are clearly stated. | We appreciate the time you spent providing helpful feedback to improve our manuscript. Below we outline the changes we have made to the manuscript to address your comments. | NA |
| R1.2 | First, the abstract would benefit from more plain language. While abbreviations like AET and PET are defined, CCAM is introduced without explanation. Removing it or adding a brief description of what CCAM refers to and why it is used would make the abstract more accessible to non-specialist readers. | We have modified the abstract to reduce complexity as suggested by the reviewer. This has included removing any reference to "CCAM", which we now term "downscaled CMIP6 models" which will make it more intelligible to readers. | DONE |
| R1.3 | The introduction is very thorough but also very long, and the study's aims only become apparent at the end. The mention of "old observational datasets" seems abrupt and lacks sufficient context. Clarifying how these datasets relate to the objectives would improve readability and make it easier to follow. | We have revised the Introduction to improve logical flow and readability. The study's aims are now introduced earlier (e.g., in the second paragraph in Introduction) as suggested. We have downsized the introduction by moving some parts to the methods to improve readability and emphasize the objectives of our contribution. | DONE |
| R1.4 | The initial visual comparison of datasets across Australia is a helpful starting point, but it would be even more informative if it were supplemented with a quantitative | We have implemented the signal-to-noise ratio analysis in order to quantify the model spread and uncertainty of the projected changes of the spatial maps. In accordance with these changes we have added the following paragraph into the methods to explain the approach (Lines 284-290): | DONE |

| | assessment of spread or uncertainty between the datasets, e.g. a map of the model spread. Specifically, identifying regions with the largest disagreement among models would highlight areas where confidence is lower and further improvement is needed. | *"We examined the signal-to-noise ratio to determine where the climate change signal emerges from the 'noise' of the model ensemble (Hawkins and Sutton, 2011). Here, we take the signal as the model ensemble average, while the noise is calculated as the standard deviation of all the projections. As we focus on end of century climate change impacts, model uncertainty is considered as noise and is expected to be the greatest source of uncertainty. Stippling is shown on ensemble mean change maps where the signal-to-noise ratio is greater than 1.0 (Chapman et al., 2024), indicating agreements among models and implying a higher degree of confidence."*

We have revised Figure 4 and Figure S2 to include the signal-to-noise ratio and so the following sentence has been added in Figure 4 Caption and Figure S2 Caption:
*"Stippling shown where the signal-to-noise ratio > 1."*

We have also included an additional figure (Figure 5) to outline the spread of the projections from all models for all emissions scenarios across Australia by the end of the century for both AET and PET. In accordance with the changes to these figures, we have added the following paragraphs to the Results (Lines 419-434):
*"For AET, there are a few areas where the signal-to-noise ratio is greater than 1, most notably along coastal eastern and northern Australia. Generally, model agreement is greater in DJF than in JJA, and greater for the high emissions scenario (SSP370) than the moderate or low emissions scenarios. By contrast, PET can be seen to have generally had a widespread model agreement according to the signal-to-noise ratio across the whole country, with a few minor exceptions. These differences relate to the very clear increases noted for PET due to increasing temperatures, which are not reflected in AET due to the majority of Australia being water limited rather than energy limited.*
*While there is a clear sign of an increase in PET across Australia by the end of the century for all the models considered across all emissions scenarios (Figure 5), the magnitude of the changes can be seen to vary among individual model members. By contrast, for AET, there is disagreement among the individual model members on the sign of the change. For example, for SSP126 while most models show a decreasing signal, there are two models which project increases. For the moderate emission scenario (SSP245) most models project decreases, whereas for the high emission scenario (SSP370), most models project increases. Even when using the same emissions scenario, the projected changes in* | |

| | | | |
|---|---|---|---|
| | | *AET can differ significantly between models, highlighting a key aspect of climate modelling uncertainty and variability in the projections."*

*And (Lines 445-449):*
*"After bias correction the model agreements have been improved for AET, particularly for SSP370 in DJF season and ANN (refer to the stippling in Figure S1 rows 1-3). For PET, bias correction also improved the consistency across models in some regions, as the signal-to-noise ratio was noted to be greater than 1 across nearly the whole country (Figure S1 rows 4-6)."* | |
| R1.5 | Adjusting the order of SSP scenarios in Figure 6 to the order used in Figures 7 and 8 would make it more intuitive to interpret the results. Also, I would include the caption entirely for Figure 8, even if it is the same as for Figure 7, so that the figure can stand on its own and readers don't have to jump between the two. | We have adjusted the order of SSPs in Figure 6 and 7 to the order used in Figures 8 and 9. Also, we have included the caption entirely for Figure 9. Note that we have updated the figure numbers after adding a new figure (Figure 5). | DONE |
| R1.6 | Lastly, regarding the drivers of change, CMIP6 models usually include LUC in their scenarios. It would be interesting to discuss if some of the changes you identify can be tied to LUC instead of CC. What do you think? | As suggested, we have included the following sentences in the discussion to explain how these changes in AET and PET are influenced by land use changes, and not just climate change (Lines 641-644):
"Regarding the drivers of change in AET and PET, some of the changes we identify can be tied to land use change, not just climate change. For example, deforestation or agricultural practices can alter surface water availability and vegetation cover, impacting AET, while changes in land surface properties (like albedo) can affect PET." | DONE |
| *Reviewer #2* | | | |
| R2.1 | The manuscript uses dynamically downscaled CMIP6 datasets as input to estimate Actual Evapotranspiration (AET) and Potential Evapotranspiration (PET) under historical and various future climate scenarios. It also employs a random forest | Thank you very much for your review and the constructive comments. We appreciate the time you have spent reading, reviewing and writing this review report. Below we outline how we have addressed each of your comments. | NA |

| | | approach to identify key driving factors influencing projected changes in AET and PET. In addition, the study evaluates multiple datasets against site observations, providing a valuable reference for selecting appropriate AET or PET products. I appreciate the authors' efforts in conducting these evaluations and projections. However, several issues should be addressed before the manuscript is suitable for publication. | | |
|---|---|---|---|---|
| R2.2 | 1. Justification of CMIP6 model selection The authors should clarify the rationale behind the selection of specific CMIP6 models and ensembles. Why were these models chosen? Do other CMIP6 models not provide the relevant variables? A brief explanation would help readers understand the basis for the selection. | We have included additional details in the methods section to justify the CMIP6 model selection (Lines 166-171): *"The ensemble of CMIP6 models chosen for downscaling in this work was selected in order to best represent the future spread in the climate change signal from all the full ensemble of CMIP6 models, while also prioritising models which were statistically independent and better able to represent the Australian climate (Trancoso et al., 2023). The climate change signal and the Kling-Gupta efficiency (KGE) skill score for historical simulations were used to select the best performing ensemble runs from different GCMs."* | DONE |
| R2.3 | 2. Consistency of projections among models The manuscript uses the mean values from multiple CMIP6 projections. However, it is unclear whether the individual models indicate consistent changing trends (e.g., all showing an increase or decrease). Are there any models that suggest an opposite direction of change, which may have been masked by averaging? This should be discussed to provide a clearer picture of the uncertainty and variability in the projections. | To assess the consistency of the projections, we have included an additional figure (Figure 5) outlining the spread of the projections from all models for all emissions scenarios across Australia by the end of the century for both AET and PET. We have also updated our spatial maps of projected changes to include the signal-to-noise ratio to highlight where the climate change signal emerges from the noise of the ensemble of climate models. In accordance with these changes, we have revised our results section (Lines 419-434): *"For AET, there are a few areas where the signal-to-noise ratio is greater than 1, most notably along coastal eastern and northern Australia. Generally, model agreement is greater in DJF than in JJA, and greater for the high emissions scenario (SSP370) than the moderate or low emissions scenarios. By contrast, PET can be seen to have generally had a widespread model agreement according to the signal-to-noise ratio across the whole country, with a few minor exceptions. These differences relate to the very clear increases* | DONE |

| | | *noted for PET due to increasing temperatures, which are not reflected in AET due to the majority of Australia being water limited rather than energy limited.* | |
|---|---|---|---|
| | | *While there is a clear sign of an increase in PET across Australia by the end of the century for all the models considered across all emissions scenarios (Figure 5), the magnitude of the changes can be seen to vary among individual model members. By contrast, for AET, there is disagreement among the individual model members on the sign of the change. For example, for SSP126 while most models show a decreasing signal, there are two models which project increases. For the moderate emission scenario (SSP245) most models project decreases, whereas for the high emission scenario (SSP370), most models project increases. Even when using the same emissions scenario, the projected changes in AET can differ significantly between models, highlighting a key aspect of climate modelling uncertainty and variability in the projections."*
*And (Lines 445-449):*
*"After bias correction the model agreements have been improved for AET, particularly for SSP370 in DJF season and ANN (refer to the stippling in Figure S1 rows 1-3). For PET, bias correction also improved the consistency across models in some regions, as the signal-to-noise ratio was noted to be greater than 1 across nearly the whole country (Figure S1 rows 4-6)."* | |
| R2.4 | 3. Definition of CCAM
The abstract mentioned CCAM without defining it. The full name should be provided upon first mention. | We have removed reference of "CCAM" in the abstract which we instead term "downscaled CMIP6 models" in order to reduce complexity. We introduce "CCAM" with its full definition in the main text, where we can explain the model without space constraints. | DONE |
| R2.5 | 4. Improving logical flow in the Introduction. The introduction could benefit from improved coherence. While the authors have evaluated both AET and PET, the transitions between topics are sometimes abrupt. For instance, around line 55, the discussion shifts from AET to PET and then back to AET, which disrupts the logical flow. Strengthening the narrative structure would enhance readability. | The Introduction has been revised to improve both logical flow and readability. We have added more information about PET (e.g., control factors and PET applications) to improve the transitions between AET and PET in Introduction. Also, we have restructured the Introduction accordingly to strengthen the narrative structure and enhance readability. | DONE |

| R2.6 | 5. Figure 1 caption clarity
The caption for Figure 1 does not explain the meaning of the solid and dashed lines, which makes it difficult to interpret the boundaries of the eight Natural Resource Management (NRM) regions. Although the regions are numbered, a clearer description of the line styles is needed. | We have included an additional sentence in the caption of Figure 1 to explain the meaning of the solid and dashed lines as suggested (Lines 147-149):
"*The solid lines represent the boundaries of the eight Natural Resource Management (NRM) regions, and dashed lines represent the boundaries for Queensland.*" | DONE |
|---|---|---|